# Effect of Organomontmorillonite-Cloisite^®^ 20A Incorporation on the Structural and Drug Release Properties of Ureasil–PEO Hybrid

**DOI:** 10.3390/pharmaceutics15010033

**Published:** 2022-12-22

**Authors:** Celso R. N. Jesus, Eduardo F. Molina, Ricardo de Oliveira, Sandra H. Pulcinelli, Celso V. Santilli

**Affiliations:** 1Instituto de Química, UNESP, Rua Professor Francisco Degni 55, Araraquara 14800-900, SP, Brazil; 2Laboratório Sol-Gel, Universidade de Franca, Av. Dr. Armando Salles Oliveira 201, Franca 14404-600, SP, Brazil

**Keywords:** ureasil-polyether, organoclay, nanocomposite, drug release

## Abstract

This paper presents the influence of the presence of a modified organoclay, Cloisite^®^ 20A (MMTA) on the structural and drug release properties of ureasil organic–inorganic hybrid. Sol–gel process was used to prepare the hybrid nanocomposites containing sodium diclofenac (DCF) at 5% wt. The effect of the amount of MMTA incorporated into the ureasil hybrid matrix was evaluated and characterized in depth by different techniques such as X-ray diffraction (XRD), small angle X-ray scattering (SAXS), differential scanning calorimetry (DSC), and swelling properties. The influence of MMTA on ureasil nanocomposites release profile was evaluated by in situ UV–vis. The diffraction patterns of the UPEO–MMTA nanocomposites showed a synergistic contribution effect that led to an intensity increase and narrowed the diffraction peaks, evidencing a crystallite PEO growth as a function of the modified nanoclay content. The interactions between polyether chains and the hydrogenated tallow of MMTA led to an easy intercalation process, as observed in UPEO–MMTA nanocomposites containing low (1% wt) or high (20% wt) nanoclay content. The waterway (channels) created in UPEO–MMTA nanocomposites contributed to a free volume increase in the swollen network compared to UPEO without MMTA. The hypothesis of the channels created after intercalation of the PEO phase in the interlayer of MMTA containing organoammonium ions corroborates with the XRD results, swelling studies by SAXS, and release assays. Furthermore, when these clay particles were dispersed in the polymeric matrix by an intercalation process, water uptake improvement was observed, with an increased amount of DCF release. The design of ureasil-MMTA nanocomposites containing modified nanoclay endows them with tunable properties; for example, swelling degree followed by amount of controlled drug release, opening the way for more versatile biomedical applications.

## 1. Introduction

The use of clay combined with organic components results in complex architectures such as those found, for example, in pigments called Maya Blue (or Prussian Blue) [1,2], prehistoric frescos [3], Chinese porcelain [4], and more. It is known that Maya Blue is based on the indigo dye molecules incorporated within the lamellae structure of palygorskite [3,5,6]. The clay–polymer conjunction opened up a wide range of possibilities to produce multifunctional systems, which fall within the “realm of organic–inorganic hybrid materials (OIH)”. In this sense, biologists, chemists, engineers, and physicists are present in this realm to develop multifunctional OIH nanocomposites. The OIH can provide potential solutions to address a series of problems, including multidrug resistance, poor drug solubility, high drug load, biocompatibility/stability, etc. By integrating organic and inorganic components, it is expected to produce new properties (chemical and physical) with unique advantages over each component separately [7,8,9,10].

The first layered silicates-based polymer nanocomposites were developed by Toyota researchers in which silicates as a nanofiller showed effective enhancements in mechanical and flame-retardant properties [11]. At that time, these materials were identified as promising nanocomposites, especially in the field of automobile industry, packaging, electronics, and coating. The application of these nanocomposites in health has received considerable attention in the last decade. In this case, the judicious choice of nanoclay (organomodified or not) could lead to the distinct mechanism and release profile of drugs [12,13,14,15]. One of the major concerns is modulating the amount of drug released by the controlled dispersion of nanofillers through a polymer matrix. It is difficult to predict whether the released amount of drug will increase, or if the barrier effect will decrease the diffusion of these molecules in an aqueous medium. Pagano et. al. [13] prepared a hybrid adhesive formulation patch based on ketoprofen intercalated into lamellar anionic clay ZnAl-hydrotalcite. In this case, the diffusion of the ketoprofen drug was dependent on the formulation of the hybrid ZnAl-hydrotalcite matrix. Thus, the main focus of the current work is to address this challenge and to demonstrate that a modified organoclay incorporated into an ureasil hybrid matrix presents distinct properties when compared with the barrier effect created in unmodified clay–polymer nanocomposites.

Ureasils are a class of OIH materials that could be used in diverse fields, such as optics [16], energy [17], health [18], and environmental [19], among others. Ureasil hybrids have been prepared by reactions between a polyetheramine and an isocyanate organosilane using sol–gel process to covalently connect the siloxane backbone through urea bridges with poly(oxyalkylene) chains [20,21,22,23]. The polyetheramine organic phase possesses flexibility and can improve dispersibility, biocompatibility, and stability. By controlling hydrolysis and condensation (sol–gel reactions) of the organosilane precursor, mechanical strength, thermal stability, and 3D insoluble networks are formed. The remarkable properties of ureasil based materials, such as rubbery, flexibility, transparency, mechanical resistance, hydrogel behavior, and insolubility in water, result from the functional groups present through the structure: urea, ether type oxygen, and silanol [24,25,26].

The physical properties of nanocomposites can be improved by the compatibility between nanoclay and polymer. The literature has already shown the striking reinforcement of polymer nanocomposites by adding small amounts of nanoclays [27,28]. Zhang et al. [29] have synthesized a variety of acrylonitrile–butadiene rubber (NBR/clay) nanocomposites that showed a good dispersion of clay layers through the rubber matrix, leading to remarkable enhancement effects to tensile strength and elongation at break of NBR/clay materials. He et al. [30] reported a novel method named gel compounding to obtain NBR/clay nanocomposites in which clays were exfoliated in water to compound with solid rubber (NBR). By using this method, the nanocomposites containing up to 20% wt of clay showed improved interfacial interaction between NBR silicate layers increasing the strength of the rubber. Furthermore, modifying nanoclays with cationic or nonionic surfactants has been an interesting strategy for tuning their compatibility with organic materials [31,32]. In this context, the incorporation of Cloisite^®^ 20A (MMTA), Gonzales, TX, USA, nanoclay (Southern Clay Products Inc., Gonzales, TX, USA) into an ureasil hybrid matrix was evaluated. For this purpose, the effect of MMTA addition (0, 1, 3, 5, 10, and 20% wt) on structural, water uptake, thermal properties, and drug delivery profile (diclofenac sodium) were studied. The organophilic MMTA is a modified montmorillonite with dimethyl dehydrogenated tallow quaternary ammonium chloride. Previously, we have shown that the presence of sodium montmorillonite (MMT) in ureasil-poly (ethylene oxide)–PEO leads to antagonistic effects with increasing MMT load into hybrid matrix [33]. The addition of MMT into ureasil–PEO induces (i) PEO crystallite size decrease, (ii) water flow barrier restricting the swelling over time, and (iii) controlled drug dissolution.

The present work aims to contribute to the field of hybrid nanocomposites for drug delivery and, more specifically, to demonstrate the easier intercalation of PEO in the organoammonium silicate layer which in turn results in an expansion of the interlayer spacing of MMTA, generating channels to flow water molecules through these spaces, contributing to an increase in both swollen network and amount of drug released as a function of time. Thanks to the combined modulation water uptake and swelling of the matrix and diffusion of drug, these nanocomposites based on ureasil–PEO containing Cloisite^®^ 20A (MMTA) can be used as novel vehicles for assisting drug transdermal delivery.

## 2. Experimental Section

### 2.1. Materials

O,O′-Bis(2-aminopropyl) polypropylene glycol-block-polyethylene glycol-block-polypropylene glycol with a molecular weight of 1900 g mol^−1^, 3-(Triethoxysilyl)propyl isocyanate (ICPTES), ethanol (CH_3_CH_2_OH), tetrahydrofuran (THF), and diclofenac sodium (DCF) were purchased from Sigma-Aldrich. Phyllosilicate Cloisite^®^ 20A (MMTA) was purchased from Southern Clay Products, Gonzales, Texas, US. All reagents were used without further purification.

### 2.2. Synthesis of the Ureasil–PEO Hybrid Precursor

The ureasil synthesis is a well-studied process that can be easily reproduced as demonstrated in the literature [34,35,36]. In general, to form an ureasil matrix, the polyether is covalently linked to a siliceous inorganic skeleton by the reaction of amino propyl groups of the functionalized PEO with 3-(isocyanatopropyl)-triethoxysilane in a molar ratio of 1:2 [34]. Therefore, these reagents were stirred together (10.4 mL of ICPTES were mixed with 40 g of PEO) in tetrahydrofuran THF (80 mL) under reflux for 24h. Following, the THF solvent was eliminated by evaporation at 60 °C to obtain a hybrid precursor containing urea bridges, see Figure 1.

### 2.3. Synthesis of the Ureasil–PEO Nanocomposites Containing Cloisite^®^20A and Sodium Diclofenac (DCF) 

To obtain the ureasil–PEO containing phyllosilicate, a desired mass of 0.015, 0.075, 0.15, or 0.30 g of MMTA was mixed with 3.0 mL of anhydrous ethanol to prepare the nanocomposites with different percentages (1, 3, 5, 10, and 20% wt with respect to the hybrid precursor’s mass) of MMTA. For better dispersion of the clay, the mixture was kept in a 30 kHz sonicator (vibracel VC 501) for 5 min. After that, 1.0 mL of each MMTA suspension was transferred to different beakers containing 1.5 g of the hybrid precursor. For the 5% wt drug-loaded samples, 0.075 g of DCF was added to the Cloisite^®^ 20A dispersion containing the hybrid precursor. These mixtures were then left under mechanical stirring for 4 h at room temperature (≈25 °C); following, the acid-catalyzed sol–gel reactions were promoted by the addition of 36 μL of HCl (2M) and 100 μL of distilled water, leading to the formation of a gel. Monolithic disks were obtained after drying in desiccators under vacuum at 70 °C for 24 h. The final materials were named as UPEO (unloaded sample) and UPEO–MMTA (loaded with Cloisite^®^ 20A).

### 2.4. Characterization

X-ray diffraction (XRD) patterns were measured with a Siemens D5000 powder diffractometer operating at 40 KV and 30 mA and equipped with a curved graphite monochromator that yields a CuKα X-ray beam (λ = 1.5405 Å). Data were collected in a 2θ angular range between 2 and 70° with step-scan of 0.02°/3 s^−1^.

Differential Scanning Calorimetry (DSC) were carried out in a TA Instrument model Q100, using ~15 mg of the ureasil xerogel. The sample was placed into a 40 mL aluminum can and heated from −90 to 350 °C at 10 °C min^−1^ under dynamic atmosphere of nitrogen gas at 70 cm^3^ min^−1^ flow rate.

The nanoscopic structures of the ureasil xerogel were studied by small-angle X-ray scattering (SAXS) at the LNLS synchrotron (Campinas, Brazil). The SAXS1 beamline is equipped with a bent Si(111) monochromator that yields a horizontally focused beam (λ = 0.1608 nm). A two-dimensional detector was used to record the X-ray intensity, I(q), as a function of the modulus of the scattering vector q = (4π/λ)sin(θ/2), with θ being the scattering angle and λ the X-ray wavelength. In the case of dried samples, the SAXS patterns were collected at 25 °C. In the case of the in situ monitoring of the swelling process, a constant flow (≈0.5 cm^3^ min^−1^) of deionized water was used for the monolithic ureasil xerogel maintained at 37 °C, and SAXS patterns were recorded every 30 s. The scattering intensity data was normalized considering the varying intensity of the direct X-ray beam, detector sensitivity, sample transmission and the parasitic scattering.

The in vitro DCF release was studied by UV–vis spectrometry, using about 0.5 g of monolithic xerogel disc immersed in 100 mL of deionized water at 37 °C. The absorption spectra data were recorded in the 190–490 nm range, using a Varian Cary 50 dual beam spectrophotometer fitted with a fiber optic coupler equipped with an immersion probe with optical path length of 5 mm. The acquisition scan rate was 300 nm min^−1^. Aqueous DCF standard solutions, at different concentrations, were used to construct a calibration curve for quantitative determination of the cumulative DCF release.

## 3. Results and Discussion

### 3.1. Structural Evaluation

The possible exfoliation/intercalation of the MMTA and the influence of the nanoclay into the ureasil–PEO hybrid structure were evaluated by XRD studies. Figure 1 shows the comparison of the XRD patterns of MMTA, UPEO, and UPEO–MMTA containing different amounts of nanoclay. The UPEO displays two narrow and well-defined peaks at 18.7° and 22.9°, characteristic of the semi-crystalline structure of PEO. Moreover, a broad diffraction peak between 15–30° is assigned to the amorphous polymer phase [22]. The diffraction patterns of the UPEO–MMTA nanocomposites showed a synergistic effect leading to an intensity increase and narrowing of the diffraction peaks as the clay content increased (see Figure 1a). This feature demonstrates an increase in the PEO crystallite size (calculated by Scherrer equation) [37,38] which can be the consequence of a heterogeneous nucleation and growth process induced by the presence of MMTA clay (Figure 1b). This effect can be associated to the presence of hydrogenated tallow groups in the MMTA (Cloisite^®^ 20A), which favor the formation of immobilized regions into the UPEO structure near the interface between the hydrophobic nanoclay platelets and the PEO hydrophilic entanglements. A distinct effect using a sodium montmorillonite (Na–MMT) loaded into the UPEO matrix, whose clay dispersion showed an antagonistic effect with a decrease in the PEO crystallite size, was reported by our research group [33].

For nanocomposites containing more than 5% wt of MMTA, the characteristic (001) peak of the nanoclay around 2θ = 3.6°, with d-spacing of 2.45 nm, was downshifted to 2.8° (5% wt MMTA) and 2.7° (10 and 20% wt MMTA), with d-spacing of 3.15 nm and 3.27 nm, respectively (see Figure 1a). This shift of the 001 peak of the MMTA clay indicates an increase in the inter-lamellae distance [39], demonstrating an intercalation of the PEO chains into the MMTA gallery.

A comparison of the XRD patterns of DCF powder, UPEO, DCF-loaded UPEO hybrid, and UPEO–MMTA–DCF nanocomposites with different quantities of MMTA are shown in Figure 1c. The incorporation of the DCF drug into UPEO induced a decrease in the diffraction peaks’ intensities at 18.7° and 22.9°, indicating a decrease in the PEO crystalline phase. This feature, consistent with the decrease in crystallinity degree reveled from DSC measurements, could result from changes in the helical conformation of the PEO crystals due to the cations coordination (Na^+^) from DCF by the ether-type oxygen of the polyether chain, enhancing the ionic dissolution of the drug by the UPEO matrix. [40] Regardless the amount of MMTA into UPEO, the presence of DCF contributes to an almost suppression of the heterogeneous growth of the PEO crystal. Regarding UPEO nanocomposites with highest amount of clay (20% wt), a notable decrease in MMTA d_001_ spacing coupled to absence of diffraction peaks of semi-crystalline PEO clearly demonstrates the intercalation of PEO chains into the MMTA gallery and the consequent loss of the UPEO matrix crystallinity (Figure 1c). It is noteworthy that the endothermic event observed by DSC coupled to the absence of PEO diffraction peak can be explained by the presence of the nanocrystalline polymeric phase [33].

SAXS curves for MMTA, UPEO hybrid, and UPEO–MMTA containing different amounts of clay are displayed in Figure 2a. The effect of DCF incorporation in SAXS curves for these samples are shown in Figure 2b. The SAXS curve of MMTA showed two diffraction peaks at q = 5.10 nm^−1^ and q = 2.55 nm^−1^, characteristic of the (002) and (001) diffraction, with a d-space of 2.46 nm, similar to the basal value observed by XRD. The UPEO SAXS curve showed a single broad peak with maxima centered at q_max_ =1.52 nm^−1^. This correlation peak, well established in the literature [19,23,34], is characteristic of the spatial correlation (ξ_d_ = 2π/q_max_) between the siloxane nodes embedded in the polymeric PEO phase, displaying an average correlation distance of 4.13 nm. Regardless the nanocomposite UPEO containing MMTA (1–20% wt), a shift to lower q values (from 5.10 nm^−1^to 3.31 nm^−1^) was observed, indicating an increase in d-space to 3.80 nm due to the intercalation of the UPEO into the clay interlayer space (Figure 2a,b). This feature could be explained by the presence of the ammonium salts between the clay platelets of Cloisite^®^ 20A, which favors the interactions with urea groups of the UPEO hybrid. Moreover, the hydrophobic interactions between polyether chains and the hydrogenated tallow of MMTA leads to an easy intercalation process, as observed in UPEO–MMTA nanocomposites containing low (1% wt) or high (20% wt) clay content. A similar behavior in SAXS curves was observed for UPEO–MMTA nanocomposites containing DCF (Figure 2b). These findings are very important and will be correlated/discussed with swelling and release assays.

### 3.2. Thermal Features

Quantitative information concerning the effect of MMTA addition and DCF loading on the crystalline and amorphous phase present in the hybrid nanocomposites were obtained from DSC analysis. Semi crystalline ureasil siloxane–PEO hybrids present characteristic thermoreversible events, such as glass transition (T_g_), crystallization (T_c_), and melting (T_m_) temperatures [23]. The heat flux change around −52 °C demonstrates the glass transition, characterized by the difference between the heat capacity of the glass and the rubber state of the amorphous phase of UPEO hybrid and nanocomposites (Figure 3a). The endothermic peak at ~33 °C corresponds to the melting of the PEO crystalline phase, which allows to calculate the crystallinity degree (D_C_) as respect to the melting enthalpy for 100% crystalline PEO (188.9 J/g) [41]. The exothermic peak observed between T_g_ and T_m_ for the nanocomposite containg DCF (Figure 3a) corresponds to the cold crystalization of PEO.

Regardless the sample’s nature (hybrid UPEO or nanocomposites containing MMTA and DCF) a unique T_g_ and melting peak was observed in the thermograms. These results suggest the absence of micro phase separation associated with the presence of a mixture of free UPEO in equilibrium with UPEO micro domains formed by interactions with organic quaternary ammonium cations of MMTA and DCF. T_g_ values of UPEO hybrid (−52.5 ± 0.7 °C) almost stayed invariant with low MMTA incorporation (1–10% wt), indicating that the amorphous bulky polymer chains remain unchanged. At highest MMTA contents (UPEO–MMTA20), T_g_ values decreased to −55.65 ± 0.5 °C, indicating a decrease in the PEO chains rigidity. The T_g_ decrease suggest that the PEO interchain interactions were affected by the organoammonium molecules present at the surface of the MMT layers. Furthermore, the crystallinity degree D_c_ of unloaded ureasil–PEO hybrid increases from 21.5% for UPEO to 25.2% for UPEO nanocomposite with 20% wt of MMTA. Therefore, the T_g_ decrease followed by the D_c_ increase demonstrates the role of the nanofillers as nucleation agents as demonstrated also by XRD results (see Figure 1b).

The addition of 5% wt of the DCF drug to the UPEO hybrid cause a significant increase in T_g_ (from −52.5 ± 0.7 °C to −49.5 ± 0.5 °C), coupled to a decrease in T_m_ (from 32.7 ± 0.4 °C to 26.0 ± 0.5 °C) and D_C_ (from 21.5 ± 0.3 °C to 18.8 ± 0.4 °C). For UPEO–MMTA nanocomposites containing the DCF drug, a similar effect on T_g_ values was observed with low MMTA content (1–5% wt). However, for UPEO–MMTA–DCF containing high clay content (10 and 20% wt), an increase in T_g_ values to −48.9 °C and −45.4 °C was observed, respectively. These last T_g_ values resemble the UPEO loaded with DCF drug. Thus, UPEO tends to (i) solubilize the DCF and (ii) intercalate in the silicate layers. In other words, interactive environments for DCF by PEO + organoammonium ion present in the hybrid matrix can be created.

The suggestion of various interaction environments in UPEO containing MMTA and DCF are corroborated by the behavior of T_m_ and D_c_ in the nanocomposites. For drug free UPEO–MMTA samples, a T_m_ decrease was observed compared with UPEO. This effect was more pronounced in the UPEO–MMTA nanocomposites containing the DCF drug. Thus, both systems (nanocomposites without and with the drug) evidence a miscibility effect on the crystallization behavior of the UPEO hybrid. The presence of the DCF drug in UPEO–MMTA nanocomposites has also a major impact in the D_C_ compared with UPEO–MMTA without the drug. Additionally, once the DCF is incorporated into UPEO–MMTA, the sharp endothermic peak at 298 °C, characteristic of the drug melting [42], was not observed (Figure 3b), demonstrating a great miscibility of DCF in all nanocomposites.

### 3.3. Swelling Behavior from SAXS Studies

The SAXS curves recorded during the swelling experiments allow us to obtain the expansion factor (Δξ/ξ_0_) calculated from the average distance between the siloxane nodes as a function of time (Figure 4). The nanoscopic dynamic of expansion of UPEO demonstrates the hydration process (water uptake) and a fast formation of a fully water-embedded hybrid matrix. Since Cloisite^®^ 20A consists of a montmorillonite modified with ammonium organic hydrogenated tallow enhancing the hydrophobicity of the clay, it was expected that the incorporation into UPEO could hinder the diffusion of water molecules through the hybrid network. However, the UPEO–MMTA nanocomposites improved the water uptake compared with the UPEO hybrid (see Figure 4d). As the MMTA content increases from 1 to 5% wt, a hydration increase by the nanocomposites was observed. This synergic effect could be correlated to the easier intercalation of PEO in the organoammonium silicate layer that expand the interlayer spacing of MMTA (see XRD results), generating channels for water molecules to flow through these spaces (see Figure 5). The waterway (channels) created in UPEO–MMTA nanocomposites contributed to an increase in the free volume of the swollen network compared with UPEO without MMTA clay. As the MMTA contents increases into UPEO, more channels were generated from intercalated structures leading to an expansion factor increase (Δξ/ξ_0_) from 50 (UPEO) to 57, 65, and 69% as a function of MMTA amount (1–5% wt). (Figure 4d). These results agree well with the DCF release results discussed below.

### 3.4. Release Assays

The cumulative release profiles from UPEO and UPEO–MMTA nanocomposites loaded with DCF (5% wt) are shown in Figure 6a. Water diffusion (hydration) in the UPEO hybrid matrix led to dissolution and fast release of DCF, reaching equilibrium after about 2h. In this short period of time, 67% of the DCF was released from the UPEO (Figure 6a black circles). Regarding UPEO–MMTA nanocomposites, a DCF release increase (at t = 200 min) as a function of clay content was observed compared with the UPEO hybrid matrix. These results agree well with the swelling behavior observed by SAXS and suggest that the intercalated polymeric PEO chains into the organoammonium silicate layer favor the water diffusion through the matrix, leading to DCF release increase. The water channels proposed for UPEO–MMTA nanocomposites (see Figure 5) correlated with an increase in the free volume of the swollen network, favors the DCF release, leading to a drug release growth at the final assay. The release profiles of UPEO–MMTA nanocomposites with lower clay content (1–5% wt) were similar in the early stage (0 < t < 60 min). In contrast, high levels of MMTA into UPEO (10 and 20% wt) provide different release profiles with significant changes in the DCF amount released at the same time (0 < t < 60 min). This justifies a higher compatibility between organomontmorillonite-modified (Cloisite^®^ 20A) with UPEO hybrid matrix which could promote a facile dispersion of PEO into the lamellae clay.

The Korsmeyer–Peppas model [43] was used to study the drug release kinetics. This model is based on Fick’s law of diffusion. The Korsmeyer–Peppas equation is given by Equation (1):(1)$$\frac{M_{t}}{M_{\infty }}=kt^{n}$$
where *M_t_*/*M_∞_* is the fractional solute release, *t* is the release time, k is a kinetic constant characteristic of the drug/polymer system, and *n* is an exponent which characterizes the mechanism of release assuming values of 0.50 and 1 for Fickian diffusion and Case II transport mechanism, respectively. Values of *n* between 0.55 and 0.90 can be regarded as an indication of superposition of both phenomena (anomalous transport).

The DCF release curves, plotted as log (*M_t_*/*M_∞_*) as a function of log *t*, from UPEO and UPEO–MMTA nanocomposites, are shown in Figure 6b. The DCF release in the final assays (t = 200 min) and the parameters obtained by the Korsmeyer–Peppas model are displayed in Table 1. The coefficient of determination (R^2^) for all samples was ~0.99, indicating that the release of DCF was well described by the Korsmeyer–Peppas model. [43] The experimental *n* value obtained for UPEO containing DCF was *n* = 0.55. This n value indicates a Fickian diffusion transport mechanism of the drug. The UPEO–MMTA nanocomposites samples showed *n* values between 0.68 and 1.0, characteristic of the anomalous transport attributed to the diffusion/zero-order rate of drug controlled by the penetration of the water-swollen front. Regarding the nanocomposite containing 5% wt of Cloisite^®^ 20A, a Case II transport was obtained (see details in Table 1).

The literature has shown the importance of nanoclay and matrix choices to prepare nanocomposites applied as drug delivery systems. Liu et al. [44] achieved a maximum water-barrier effect using clay–polymer nanocomposites based on exfoliated montmorillonite–Eudragit containing 10% of Cloisite^®^ 20. In another work, Adrover et al. [45] demonstrated that the presence of laponite in a gellant gum bead formulation increases the drug entrapment efficiency reaching a slow-down release kinetics of drugs using a gastric environment. In general, the addition of clays in polymeric matrices leads to a barrier effect, decreasing the diffusion of the drug into the aqueous medium. In previous work [28], we showed that the presence of sodium montmorillonite (MMT) in ureasil-poly (ethylene oxide)–PEO leads to water flow barrier, restricting the swelling over time and decreasing the amount of DCF drug released. Thus, only a physical barrier was observed by the dispersion of exfoliated MMT clay at ureasil organic–inorganic matrix.

In this work, for the first time, we highlight a synergic effect between an ureasil UPEO hybrid matrix and a organomontmorillonite-modified (Cloisite^®^ 20A), prepared by sol–gel process at mild temperatures. The hypothesis of the created channels after intercalation of PEO phase in the interlayer of MMTA containing organoammonium ions is corroborated by the XRD results, swelling studies by SAXS, and release assays. Furthermore, when these clay particles were dispersed in the polymeric matrix by an intercalation process, the water uptake was improved with a consequent increase in the DCF amount release. Based on the comparisons above, this work justifies the change in nonmodified and organomodified montmorillonite nanoclay dispersed into an ureasil hybrid matrix, which leads to very distinct properties to the nanocomposites, such as swelling, crystalline degree of polymer, and diffusion of the drug molecules (decreasing or augmenting the amount released). Hence, the current strategy to use Cloisite^®^ 20A ureasil hybrid presents an impact on the health field to augment the amount of drug released as a function of the nanoclay incorporated. This enhancement in these properties may present benefits depending on the intended treatment.

## 4. Conclusions

A controlled delivery system (diclofenac sodium) based on an ureasil–PEO hybrid containing modified montmorillonite with dimethyl dehydrogenated tallow quaternary ammonium chloride (Cloisite^®^ 20A—MMTA) was designed in this study. The crystallite size of PEO increased as a function of MMTA content, as observed by XRD results. Our experimental data indicate that ureasil–PEO hybrid can be easily loaded with a range of MMTA nanoclay amounts varying between 1–20% wt, showing striking synergic effects such as a waterway flux (channels) created by the intercalation of PEO chains into the organoammonium silicate layer. These channels enable the improvement of both the water swelling degree of ureasil–PEO-MMTA nanocomposites and the amount of diclofenac drug released as a function of time. The enhancement of water uptake and DCF dissolution by ureasil–PEO nanocomposites showed that the amount of drug could be tuned by varying the MMTA loaded into the hybrid matrix, which facilitates the diffusion of the drug in the aqueous medium. It is believed that the ureasil nanocomposites containing modified nanoclays can be extended to different types of active molecules (anticancer drugs, enzymes, RNA-DNA molecules, and more), showing tunable controlled release by the choice of nanoclay (unmodified or modified) loaded into the ureasil matrices.

## Data Availability

Not applicable.

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
