# Peer review of "Effect of Organomontmorillonite-Cloisite® 20A Incorporation on the Structural and Drug Release Properties of Ureasil–PEO Hybrid"

_pharmaceutics, 2022, doi:10.3390/pharmaceutics15010033_

Round 1
Reviewer 1 Report
The paper presents some interesting work, but the work is not clearly placed in context with the extensive prior work with nanoclays.
1) The Introduction needs to clearly explain the prior art in this area and the significance of this work in advancing knowledge in this area.
2) The discussion of the results should clearly present how the new results are similar to or different from results from prior work.
3) Please check the paper for errors in notation (e.g., Cloisite is a tradename), capitalization, and grammar.
Author Response
Dear Carina Li,
Assistant Editor, Pharmaceutics
Date: 01 december 2022
Manuscript number: pharmaceutics-2040939
Title: "Effect of organomontmorillonite-Cloisite® 20A incorporation on the structural and drug release properties of ureasil-PEO hybrid”
Thank you for your attention regarding our manuscript. The reviewer’s comments were very important and certainly helped in improving the quality of the paper. Please, find enclosed the revised version of the manuscript (marked up using the “Track Changes” word format) as well as the answer to the reviewer’s comments.
With sincere regards,
ANSWER TO THE REVIWER’S COMMENTS
Reviewer # 1
General Comments. The paper presents some interesting work, but the work is not clearly placed in context with the extensive prior work with nanoclays.
- R. Thanks to the reviewer for this comment. We have compared our work with the previous paper about nanoclay-ureasil to clearly demonstrate the differences (results) when a modified organic nanoclay incorporated into ureasil hybrid matrix was used. The comparison can be find in the revised version of the manuscript (see in introduction, page 2, line 107; and results, page 14, line 620).
Comment 1. The Introduction needs to clearly explain the prior art in this area and the significance of this work in advancing knowledge in this area.
- R. Thanks to the reviewer for this comment. In the revised version of the manuscript, we highlighted the importance of the work in pharmaceutic area suggesting the possibility to use this class of ureasil xerogel as transdermal drug delivery systems by modulating the swelling of the matrix followed by the drug diffusion (see in introduction, page 2, line 117).
Comment 2. The discussion of the results should clearly present how the new results are similar to or different from results from prior work.
- R. Thanks to the reviewer for this comment. We agree with the comment, and in the revised version of the manuscript we compared our work with the previous paper about nanoclay-ureasil to clearly demonstrated the differences (results) when a modified organoclay incorporated into ureasil hybrid matrix was used (see in results, page 14, line 620).
Comment 3. Please check the paper for errors in notation (e.g., Cloisite is a tradename), capitalization, and grammar..
- R. We would like to thank the Reviewer for this suggestion. The errors in notation (e.g., Cloisite is a tradename), capitalization, and grammar were double-checked in the revised version of the manuscript.

Reviewer 2 Report
1. Introduction: 'The literature has already shown the striking reinforcement of polymer nanocomposites by adding small amounts of nanoclays. ' Both rubbers and plastics should be included since reinforcement is especially important for rubbers. So the authors are highly suggested to cite the following references about clay reinforcement for rubbers: (1) Polymer Testing, 2022. 110: 107565. (2) Composites Part B: Engineering, 2019. 167: 356-361.
2. Figure 2: why logarithmic coordinates was used in this figures?
3. More discussion should be added about the DSC curves in Figure 3.
4. What are the connections between the water uptake and drug release?
5. What role was the nanoclay play in improving the performance? Please demonstrate it clear.
Author Response
Dear Carina Li,
Assistant Editor, Pharmaceutics
Date: 01 december 2022
Manuscript number: pharmaceutics-2040939
Title: "Effect of organomontmorillonite-Cloisite® 20A incorporation on the structural and drug release properties of ureasil-PEO hybrid”
Thank you for your attention regarding our manuscript. The reviewer’s comments were very important and certainly helped in improving the quality of the paper. Please, find enclosed the revised version of the manuscript (marked up using the “Track Changes” word format) as well as the answer to the reviewer’s comments.
With sincere regards,
ANSWER TO THE REVIWER’S COMMENTS
Reviewer # 2
Comment 1 . Introduction: 'The literature has already shown the striking reinforcement of polymer nanocomposites by adding small amounts of nanoclays. ' Both rubbers and plastics should be included since reinforcement is especially important for rubbers. So the authors are highly suggested to cite the following references about clay reinforcement for rubbers: (1) Polymer Testing, 2022. 110: 107565. (2) Composites Part B: Engineering, 2019. 167: 356-361.
- We would like to thank the Reviewer for this suggestion. The cited texts were inserted in the revised version of the manuscript and helped to improve the importance of small amounts of nanoclays as striking reinforcement of polymer nanocomposites (see details in introduction).
Comment 2. Figure 2: why logarithmic coordinates was used in this figures?
- R. Thanks to the reviewer for this comment. We used a logarithmic scale to better show the scattering peak in the SAXS curve and to help later data analysis.
Comment 3. More discussion should be added about the DSC curves in Figure 3.
- We would like to thank the Reviewer for this suggestion. We inserted more detailed discussion about DSC results in the revised manuscript version. All the parameters such as glass transition, crystallization and melting temperatures concerning the PEO crystalline phase were discussed (see in results, page 10).
Comment 4. What are the connections between the water uptake and drug release?
- R. Thanks to the reviewer for this comment. The UPEO-MMTA nanocomposites have the water uptake improved as compared with the pristine UPEO hybrid, because the MMTA content increases the hydration process of the nanocomposites . The DCF release is directly related to the hydration process, due to the solubility and affinity of DCF to water molecules. The waterway created in UPEO-MMTA nanocomposites contributed to an increase of the free volume of the swollen network compared with the MMTA clay free UPEO. The increase of MMTA content into UPEO leads to an expansion and a swollen front, which improves the dissolution and diffusion of DCF molecules in water as a function of time. These correlations/discussion are presented in the manuscript (section 3.3. Swelling behavior from SAXS studies and 3.4. Release assays)
Comment 5. What role was the nanoclay play in improving the performance? Please demonstrate it clear.
- R. Thanks to the reviewer for this comment. The water uptake of UPEO-MMTA nanocomposites was improved as compared with the UPEO hybrid. This synergic effect (improving performance corresponding to expansion factor of the swollen front) could be correlated to the easier intercalation of PEO in the organoammoniium-silicate layer that caused the expansion of the interlayer spacing of MMTA, generating channels for water molecules to flow through these spaces. The waterway created in UPEO-MMTA nanocomposites contributed to an increase of the free volume of the swollen network compared with MMTA clay free UPEO. As the content of MMTA into UPEO increases more channels are generated from intercalated structures leading to an expansion factor increase from 50 % (UPEO) to 57%, 65 % and 69% as a function of MMTA amount (1 – 5% wt). These results agree well with the DCF release assays shown in the manuscript. (section 3.3. Swelling behavior from SAXS studies and 3.4. Release assays)
